# Power-Delay-Profile-Based MMSE Channel Estimations for OFDM Systems

**Sungmook Lim** [1,2], **Hanho Wang** [3,*] **and Kyunbyoung Ko** [1,2,*]

1 Department of IT-Energy Convergence (BK21 FOUR), Korea National University of Transportation, 50 Daehak-ro, Chungju-si 27469, Republic of Korea
2 Department of Electronics Engineering, Korea National University of Transportation, 50 Daehak-ro, Chungju-si 27469, Republic of Korea
3 Department of Smart Information and Telecommunication Engineering, Sangmyung University, 31, Sangmyeongdae-gil, Dongnam-gu, Cheonan-si 31066, Republic of Korea
* Correspondence: hhwang@smu.ac.kr (H.W.); kbko@ut.ac.kr (K.K.)

**Abstract:** This paper deals with the problem of the channel estimation of orthogonal frequency division multiplexing (OFDM) signals transmitted through a time-varying fading channel. We show that an accurate power delay profile (PDP) estimator can be obtained by utilizing the cyclic redundancy induced by a cyclic prefix (CP), which can be applicable to OFDM systems with insufficient pilot or training symbols (IEEE 802.11p/WAVE system), and then a least mean square error (MMSE) channel estimation scheme can be obtained in a manner based on the estimated PDP. The simulation results highlight the benefit of the proposed methods compared with the state-of-the-art standard and three achievable performance bounds.

**Keywords:** PDP; OFDM; CP; MMSE





## 1. Introduction

Many studies on cooperative intelligent transportation systems (C-ITS) for actively responding to traffic conditions through a real-time mutual communication with surrounding vehicles and infrastructure while vehicles are moving have been widely discussed. The IEEE 802.11p/Wireless Access in Vehicular Environments (WAVE) standard was developed to support vehicle wireless (i.e., vehicle-to-everything (V2X)) communication, in which the MAC and PHY of WLAN were defined [1]. Note that IEEE 802.11p is a modification of the frequency bandwidth of the IEEE 802.11a standard from 20 MHz to 10 MHz [1]. This means that channel estimation (CE) in IEEE 802.11p V2X communication systems can be a difficult task, usually due to the high mobility and insufficient number of pilots in orthogonal frequency division multiplexing (OFDM) systems [1–12].

Since the minimum mean square error (MMSE)-CE scheme for OFDM systems shows excellent channel estimation performance, extensive research related to this has been conducted [2–8]. There are two kinds of MMSE scheme: one is the pilot symbol assisted scheme requiring pilot or training symbols [2–6] and the other is the decision-directed scheme utilizing the constructed data pilots [7,8]. In [2], the maximum likelihood (ML) and the MMSE schemes were compared as the channel estimators based on pilot-aided OFDM systems. The author in [3] investigated pilot-symbol-aided parameter estimation for OFDM systems. The work in [4] dealt MMSE-CE based on power delay profile approximation. In [5], low-complexity windowed discrete Fourier transform (DFT)-based MMSE channel estimators were proposed and analyzed. An adaptive MMSE-CE was addressed in [6] related to maximum access delay time estimation. Notice that the works of [2–6] cannot guarantee a good CE performance for OFDM systems with insufficient pilot or training symbols.

Correspondingly, various improved channel estimation technologies for IEEE 802.11p systems have been developed in order to solve the problem of insufficient pilot symbols in OFDM systems and so as to accurately estimate time-varying channels [7–12]. For example, spectral temporal averaging (STA), construct data pilot (CDP) [9], time–domain reliable-test frequency–domain interpolation (TRFI) [10], weighted sum using update matrix (WSUM) [11], and MMSE channel estimation schemes have been proposed [7,8]. The authors in [12] presented a state feedback decision algorithm for data pilot-aided channel estimation in the iterative channel estimation and decoding methods. As shown in [11], a WSUM scheme can be regarded as a weighted averaging scheme in a frequency domain when it is compared with a STA scheme. The authors in [8] presented an adaptive mode-switching method between channel estimation schemes based on the MMSE technique. The MMSE channel estimation schemes in [7,8] have structures in which a correlation matrix is obtained by accumulating each OFDM symbol in a packet, and a matrix inversion related to the updated correlation matrix is performed for every OFDM symbol in a packet. Despite the high complexity of the MMSE schemes in [7,8], they do not provide satisfactory performances in a higher speed and more frequency selective channel environment. This is caused by the fact that it is insufficient to estimate the correlation matrix of a channel using only OFDM symbols in one packet.

The works in [4,13,14] dealt with the delay spread estimation based on training symbols and the SNR estimation based on the preamble or pilot symbols for OFDM systems. Recently, the non-data-aided (NDA) method for noise variance estimation was proposed in [15]. Notice that the works in [4,13,14] cannot be used to estimate the power delay profile (PDP) for the case of having insufficient pilot or training symbols. Moreover, the works in [2,5,7,8] did not address the PDP estimation issue, which can be applied at the MMSE channel estimation for OFDM systems. The work in [4] presented the approximated PDP estimation method based on pilot symbols and the MMSE channel estimation scheme for OFDM systems. The authors in [16] presented the PDP estimation methods based on pilot symbols for multiple-input multiple-output (MIMO)-OFDM systems. Nevertheless, the methods in [4,6,16] cannot be applied to the case of having insufficient pilot or training symbols.

In this paper, we employ a technique for obtaining the correlation matrix of channels that are not related to the instantaneous channel estimation for OFDM symbols in a packet. By doing so, the inverse matrix operation in the MMSE scheme can be performed only once per packet. The authors in [17] presented the noise variance and PDP estimators for OFDM systems by taking advantage of the periodic redundancy induced by the CP. The authors in [18] showed the NDA signal-to-noise ratio (SNR) estimation of the OFDM signals transmitted through unknown multipath fading channel without a subjective choice of a threshold level [17]. Recently, the authors in [19] proposed an improved PDP estimation scheme as an approximated ML-type method. In order to apply for MMSE channel estimation, three types of PDP estimators are considered as follows:

- 'Method 1': reference [17] (with a threshold level)
- 'Method 2': reference [18] & Modification (without a threshold level)
- 'Method 3': reference [19] (without a threshold level)

Notice that 'Modification' in 'Method 2' relate to reducing the amount of computation required for PDP estimation. The performance of the MMSE channel estimation scheme to which the three types of PDP estimators are applied is verified through the simulation on IEEE 802.11p/WAVE systems. For error rate performance comparison, we present three performance bounds of 'Perfect CE', 'Ideal $-$ $\mathbf{R}_{hh}$', and 'Ideal $-$ $\mathbf{p}_{L_{\max}}$', which show the limit of the time-varying channel, the practical channel where a path correlation exists, and the limit in the case where there is no path correlation in the practical channel, respectively. Through simulations considering the correlated channel matrix, it is confirmed that the performance limitations and superiority of the proposed methods are verified.

The remainder of this paper is organized as follows. Section 2 describes the discrete signal model for OFDM systems. Section 3 presents three types of PDP estimation scheme.

The PDP based MMSE channel estimation schemes are described in Section 4. Section 5 shows the simulation results, and concluding remarks are given in Section 6.

## 2. Discrete Signal Model for OFDM Systems

In OFDM systems, source data are grouped and mapped into $N$ modulated symbols $X_m(k)|_{k=0}^{N-1}$, where $E\{|X_m(k)|^2\} = 1$, and $E\{\cdot\}$ denotes the expectation. Then, by inverse discrete Fourier transform (IDFT) on $N$ parallel subcarriers, the transmitted time–domain signal of the $n$th sample for the $m$th OFDM symbol can be written as

$$x_m(n) = \sqrt{\frac{E_s}{N}} \sum_{k=0}^{N-1} X_m(k)e^{j2\pi kn/N}, \tag{1}$$

where $n \in \{0, 1, \cdots, N-1\}$, and $E_s$ is the signal power [20–22].

The guard interval is inserted to prevent interference between OFDM symbols and includes a cyclic prefix (CP) that replicates the end of the IDFT output sample. When $N_g$ is the number of guard interval samples, it is assumed to be larger than the delay spread of the multipath fading channel. The signal is transmitted over the multipath fading channel, and its low-pass channel impulse response can be expressed as

$$h(t; \tau) = \sum_{l=0}^{L-1} h_l(t)\delta(\tau - \tau_l) , \tag{2}$$

where $t$, $\tau$, $\delta(\cdot)$, $\tau_l$, and $L$ are the time, the delay, a Dirac delta function, the propagation delay of the $l$th path, and the number of multipaths, respectively [20–22]. The correlation relationship between the paths can be expressed by the wide-sense stationary uncorrelated scattering (WSSUS) model [21–23]. This model assumes that the paths are uncorrelated, and the correlation property of the channel is stationary.

When we remove CP samples, the received signal can be presented as

$$y_m(n) = \sum_{l=0}^{L-1} h_{l,m}(n)x_m((n-d_l)_N) + w_m(n) , \tag{3}$$

where $(\cdot)_N$ represents a cyclic shift in the base of $N$, $w_m(n) \sim \mathcal{N}(0, \sigma^2)$, which is an Additive White Gaussian Noise (AWGN), $h_{l,m}(n) = h_l(t)|_{t=(m(N_g+N)+n)T_s}$ is the $l$th path channel gain of the $n$th sample for the $m$th OFDM symbol, and $d_l = \lfloor \tau_l/T_s \rfloor$ is the delay normalized by the sampling time $T_s$ [22]. For simplicity, we round $d_l$ to an integer without considering leakage. However, the correlation approach in this paper may also be extended to fractional $d_l$ [17].

When we assume the perfect synchronization with $d_0 = 0$, and that the channel is time-invariant within two consecutive OFDM symbols, indexes $m$ and $(n)$ in $h_{l,m}(n)$ from (3) can be omitted as $h_{l,m}(n) \to h_l$. At the border between two OFDM symbols, the received signal samples for $-N_g \leq n < 0$ can be expressed as

$$\begin{aligned} y_m(n) \quad &= \sum_{l=0}^{L-1} h_l x_{m-1}(N+n-d_l)U(d_l-n) \\ &+ \sum_{l=0}^{L-1} h_l x_m(n-d_l)U(n-d_l) + w_m(n) \end{aligned} \tag{4}$$

where $h_l \sim \mathcal{N}(0, \sigma_l^2)$, $\sigma_h^2 = \sum_{l=0}^{L-1} \sigma_l^2 = \sum_{l=0}^{L-1} |h_l|^2$, and $U(\cdot)$ is the unit step function [17]. When we define the maximum number of paths including zero channel gain path as

$$L_{\max} = \max\{d_l\} + 1, \tag{5}$$

the maximum access delay time, normalized by $T_s$, can be written as

$$d_{\max} = \max\{d_l\} = L_{\max} - 1. \qquad (6)$$

The correlation between each received signal over CP duration and its corresponding sample at the end of the OFDM symbol can thus be expressed as [17,24]

$$E\{y_m(-k)y_m^*(N-k)\} = \begin{cases} \sigma_h^2, & 0 < k \le N_g - d_{L-1} \\ \sum_{l=0}^{L-1} \sigma_l^2 U\left(N_g - k - d_l\right), & N_g - d_{L-1} < k \le N_g - d_0, \\ 0, & N_g - d_0 < k \le N_g \end{cases} \qquad (7)$$

where $k = 1, \cdots, N_g$. Note that the expectation in (7) is taken with regard to both $\{h_l\}$ and $\{x_m(n)\}$. When $L$ is large, $y_m(n)|_{n=0}^{N-1-N_g}$ can be approximated as the complex Gaussian by using the central limited theorem, and the probability density function (PDF) can be presented as [17,24]

$$f(y_m(n)) = \frac{1}{\pi(\sigma_h^2 + \sigma^2)} \exp\left(-\frac{|y_m(n)|^2}{\sigma_h^2 + \sigma^2}\right). \qquad (8)$$

Samples $y_m(-k)$ and $y_m(N-k)$ are jointly Gaussian with the PDF of

$$f(y_m(-k), y_m(N-k)) \\ = \frac{\exp\left(-\frac{|y_m(-k)|^2 + |y_m(N-k)|^2 - 2\rho_k \Re\{y_m(-k)y_m^*(N-k)\}}{\sigma_h^2 + \sigma^2}\right)}{\pi^2(\sigma_h^2 + \sigma^2)(1 - \rho_k^2)} \qquad (9)$$

where

$$\rho_k = \frac{|E\{y_m(-k)y_m^*(N-k)\}|}{\sqrt{E\{|y_m(-k)|^2\}E\{|y_m(N-k)|^2\}}} = \frac{1}{\sigma_h^2 + \sigma^2} \sum_{l=0}^{L-1} \sigma_l^2 U(N_g - k - d_l). \qquad (10)$$

Notice that $0 < \rho_k < 1$ and $\rho_k \ge \rho_{k+1}$ (i.e., $\rho_k$ is a non-increasing value in proportion to $k$).

## 3. Power Delay Profile Estimation

Under the perfect synchronization at reception and a time-invariant channel over an OFDM symbol time, the $N_g$ noise variance estimators can be written as

$$J(u) = \hat{\sigma}_u^2 = \frac{1}{2M(N_g - (u-1))} \sum_{m=1}^{M} \sum_{k=u}^{N_g} |y_m(N-k) - y_m(-k)|^2, \qquad (11)$$

where $u \in \{1, 2, \cdots, N_g\}$, and $M$ denotes the number of OFDM symbols in the observation window [18]. Under the given environment, Figures 1 and 2 show the normalized mean square errors (NMSEs) for $N_g$ noise variance estimators from (11). It is shown that the estimator with $u = L_{\max}$ results in the smallest NMSE [18,19].

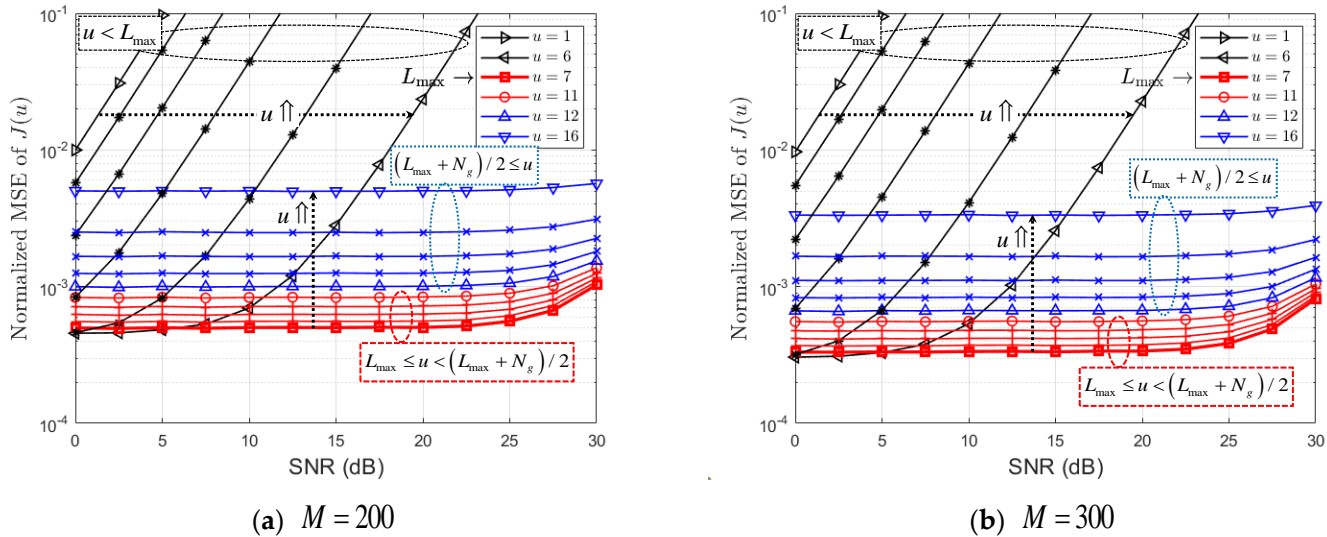

**Figure 1.** NMSE of $J(u)$ (Street Crossing NLOS, $M \in \{200, 300\}$).

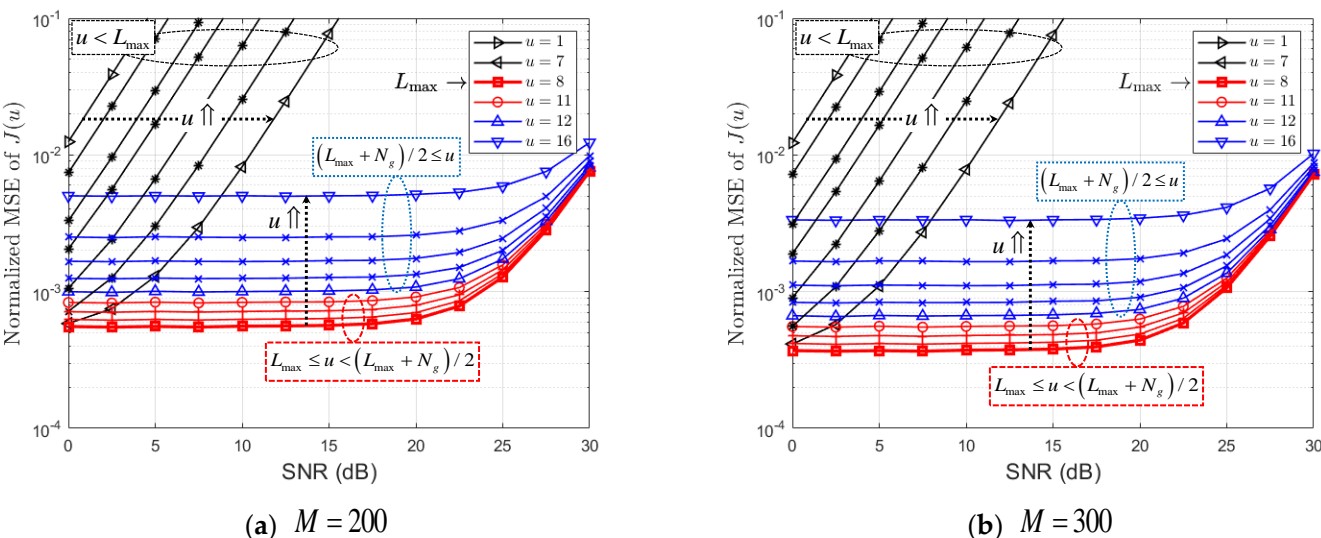

**Figure 2.** NMSE of $J(u)$ (Highway NLOS, $M \in \{200, 300\}$).

### 3.1. Method 1: [17]

Define $\mathbf{y} = [y_1(-N_g), y_1(-N_g + 1), \cdots, y_M(N-1)]$, $\mathbf{p} = [\sigma_0^2, \cdots, \sigma_{L-1}^2]$, and $\mathbf{d} = [d_0, \cdots, d_{L-1}]$. From (8) and (9), and the fact that $M$ OFDM symbols are mutually independent, the log likelihood function of $\mathbf{y}$ conditioned on $\sigma^2$, $\mathbf{p}$, and $\mathbf{d}$ can be written as [17,19]

$$
\begin{aligned}
&\Lambda\left(\mathbf{y} | \sigma^2, \mathbf{p}, \mathbf{d}\right) \\
&= \sum_{m=1}^{M} \log\left(\prod_{k=1}^{N_g} f\left(y_m(-k), y_m(N-k)\right) \prod_{k=0}^{N-1-N_g} f\left(y_m(k)\right)\right) \\
&= -M\left(\sum_{k=1}^{N_g}\left[\frac{a_k - 2\rho_k b_k}{c(1-\rho_k^2)} + \log\left(\pi^2 c\left(1-\rho_k^2\right)\right)\right] + \sum_{k=0}^{N-1-N_g} \frac{g_k}{c} + \log(\pi c)\right)
\end{aligned}
$$
(12)

where

$$a_k = \frac{1}{M}\left(\sum_{m=1}^{M}|y_m(-k)|^2 + |y_m(N-k)|^2\right)$$
$$b_k = \frac{1}{M}\sum_{m=1}^{M}\Re\{y_m(-k)y_m^*(N-k)\}^2$$
$$g_k = \frac{1}{M}\sum_{m=1}^{M}|y_m(k)|^2$$
$$c = \sigma_h^2 + \sigma^2$$

(13)

The authors of [17] showed a suboptimal way for the joint parameters' estimation from (12). At first, we can estimate $c$ in (12) as

$$\hat{c} = \frac{1}{N-N_g}\sum_{k=0}^{N-1-N_g}g_k = \frac{1}{(N-N_g)M}\sum_{k=0}^{N-1-N_g}\sum_{m=1}^{M}|y_m(k)|^2 \ ,$$

(14)

which is the time average estimation of $\sigma_h^2 + \sigma^2$. Substituting $\hat{c}$ back into the first summation from (12) and maximizing $\rho_k$ individually, we obtain the estimate for $\rho_k$ as the real root of the equation

$$\hat{c}\rho_k^3 - b_k\rho_k^2 + (a_k - \hat{c})\rho_k - b_k = 0.$$

(15)

By letting $\{\hat{\rho}_k\}_{k=1}^{N_g}$ be the real roots of $N_g$ cubic equations from (15), the estimated path power can be written temporally as

$$\hat{p}_0 = \hat{\sigma}_0^2 = \hat{\rho}_{N_g}/\hat{c}$$
$$\hat{p}_k|_{k=1}^{N_g-1} = \hat{\sigma}_k^2 = \begin{cases} \left(\hat{\rho}_{N_g-k} - \hat{\rho}_{N_g-k+1}\right)/\hat{c} & \text{if} & \hat{\rho}_{N_g-k} > \hat{\rho}_{N_g-k+1} \\ 0 & \text{else} \end{cases}$$

(16)

In [17], the authors suggested the $\hat{L}_{\max}$ estimation scheme based on a threshold value $\alpha$. If $\hat{p}_k(=\hat{\sigma}_k^2) > \alpha\hat{c}$, it is identified as a path having the estimated path power of $\hat{p}_k$ and the estimated delay time of $k$. They denoted the maximum delay time as $\hat{d}_{\max} = \max\{k\}$ satisfying $\hat{p}_k(=\hat{\sigma}_k^2) > \alpha\hat{c}$. Consequently, the estimated values can be presented for $k = 0, 1, \cdots, N_g - 1$ as follows

$$\hat{p}_k = \hat{p}_k U(\hat{p}_k - \alpha\hat{c})$$
$$\hat{d}_k = k U(\hat{p}_k - \alpha\hat{c})$$
$$\hat{d}_{\max} = \max\{\hat{d}_k\}$$

(17)

From (5) and (17), we can obtain $\hat{L}_{\max} = \hat{d}_{\max} + 1$, and thus, the estimated noise variance can be expressed as $\hat{\sigma}^2 = J(\hat{L}_{\max})$ from (11).

### 3.2. Method 2: [18] & Modification

As mentioned in [18], 'Method 1' of [17] has the major disadvantage of being based on a threshold level chosen arbitrarily. To overcome this limitation, the authors in [18] proposed a method inspired by the maximum likelihood estimation. From (11), $J(u)$ can be expressed as

$$J(u) = \left(1 - \frac{1}{N_g - (u-1)}\right)J(u+1) + \epsilon(u),$$

(18)

where $\epsilon(u)$ can be modeled as is a random variable that follows a chi square distribution for $d_{\max} \leq u < N_g$. This distribution can be simplified, for large $M$, as

$$\epsilon(u) \sim \mathcal{N}\left(\frac{\sigma^2}{N_g - (u-1)}, \frac{\sigma^4}{M(N_g - (u-1))^2}\right).$$

(19)

Then, $L_{\max}$ can be estimated using the likelihood function $f(\mathbf{X}_u|L_{\max} = u)$ with the observation variables defined as $\mathbf{X}_u = [\epsilon(u), \epsilon(u+1), \cdots, \epsilon(N_g - 1)]$. When the different $\epsilon(u)$ is assumed to be independent, $\hat{L}_{\max}$ can be obtained by

$$\hat{L}_{\max} = \underset{u}{\operatorname{argmax}} \left[ \prod_{m=u}^{N_g - 1} f(\epsilon(u)|L_{\max} = u) \right]^{/1(N_g - u)}, \tag{20}$$

where $1 \leq u < N_g$, and $f(\epsilon(u)|L_{\max} = u)$ is computed from (19) with the approximation that $\sigma^2 \simeq J(u)$. Note that, because the observations $\mathbf{X}_u$ are of variable lengths, (20) is defined as an average likelihood that is the geometric mean of the individual likelihood elements [18]. From (20), the estimated noise variance and the maximum delay time can be obtained as $\hat{\sigma}^2 = J(\hat{L}_{\max})$ and $\hat{d}_{\max} = (\hat{L}_{\max} - 1)$, respectively.

Notice that, unlike in [17], there is no need to solve $N_g$ cubic equations, because $\hat{L}_{\max}$ is determined first regardless of the threshold level. From (10), we can easily find that $\{\rho_k\}_{k=1}^{N_g - d_{\max}}$ are same. Therefore, $\{\hat{\rho}_k\}_{k=1}^{N_g - \hat{d}_{\max}}$ can be obtained as the real root of the single equation

$$\hat{c}\rho^3 - b\rho^2 + (a - \hat{c})\rho - b = 0, \tag{21}$$

where $a = \sum_{k=1}^{N_g - \hat{d}_{\max}} a_k / \left(N_g - \hat{d}_{\max}\right)$ and $b = \sum_{k=1}^{N_g - \hat{d}_{\max}} b_k / \left(N_g - \hat{d}_{\max}\right)$. Then, $\{\hat{\rho}_k\}_{N_g - \hat{d}_{\max} + 1}^{N_g}$ can be obtained by solving (15). This means that we need $\hat{L}_{\max} \left(= \hat{d}_{\max} + 1\right)$ cubic equations' real roots. Similar to (16), we can obtain the estimated path power as

$$\begin{aligned} \hat{p}_0 &= \hat{\sigma}_0^2 = \hat{\rho}_{N_g}/\hat{c} \\ \hat{p}_k|_{k=1}^{\hat{L}_{\max} - 1} &= \hat{\sigma}_k^2 = \begin{cases} \left(\hat{\rho}_{N_g - k} - \hat{\rho}_{N_g - k + 1}\right)/\hat{c} & \text{if} \quad \hat{\rho}_{N_g - k} > \hat{\rho}_{N_g - k + 1} \\ 0 & \text{else} \end{cases} \\ \hat{p}_k|_{k=\hat{L}_{\max}}^{N_g} &= \hat{\sigma}_k^2 = 0 \end{aligned} \tag{22}$$

### 3.3. Method 3: [19]

Table 1 shows the algorithm to estimate $d_{\max}$ in [19]. As mentioned in [18], Cui et al. suggested an estimator in [17], but it has the major disadvantage of being based on a threshold level chosen arbitrarily. A threshold level can be dependent on SNR. Therefore, the authors in [19] proposed an $L_{\max}$ estimation algorithm with robust characteristics in all SNR regions as a way to find an $\hat{L}_{\max}$ that maximizes the log likelihood function from (12) without a threshold value. By ignoring constant term in (12), we can represent (12), from $\{\rho_u(k)\}_{k=1}^{N_g}$ in Table 1, (13), and (14), as

$$\Lambda_p\left(\mathbf{y}, \{\rho_u(k)\}_{k=1}^{N_g} \middle| L_{\max} = u\right) = -M\left(\sum_{k=1}^{N_g}\left[\frac{a_k - 2\rho_u(k)b_k}{\hat{c}(1 - \rho_u^2(k))} + \log\left(1 - \rho_u^2(k)\right)\right]\right), \tag{23}$$

and then, $\hat{L}_{\max}$ can be obtained as in [19]

$$\hat{L}_{\max} = \underset{u}{\operatorname{argmax}}\left[\Lambda_p\left(\mathbf{y}, \{\rho_u(k)\}_{k=1}^{N_g} \middle| L_{\max} = u\right)\right]. \tag{24}$$

From (24), the estimated noise variance, the estimated maximum delay time, and the estimated PDP can be expressed as $\hat{\sigma}^2 = J(\hat{L}_{\max})$, $\hat{d}_{\max} = (\hat{L}_{\max} - 1)$, and $\mathbf{p}_{\hat{L}_{\max}} = \mathbf{p}_u|_{u=\hat{L}_{\max}}$ in Table 1, respectively. Table 2 shows the steps and comparison of three methods.

**Table 1.** Algorithm used to estimate $d_{\max}$ in [19].

| Algorithm | Comments |
|---|---|
| for $u = 1 : N_g$ | |
| $\quad \mathbf{p}_u = \mathbf{0}_{1 \times N_g}$ | $1 \times N_g$ zero row vector |
| $\quad p_u(k)\big|_{k=0}^{u-1} = \hat{p}_k = \hat{\sigma}_k^2$ | $u$ paths power selection from (16) |
| $\quad \sigma^2 = J(u)$ | $u$th estimated noise power from (11) |
| $\quad \mathbf{p}_u = \mathbf{p}_u \times (\hat{c} - \sigma^2) / \sum_{k=0}^{u-1} \hat{p}_k$ | channel power normalization |
| $\quad \rho_u(k)\big|_{k=1}^{N_g} = \sum_{l=0}^{N_g-k} p_u(l)/\hat{c}$ | $u$th $\rho$ calculation |
| end for | |

**Table 2.** Steps and Comparison of Three Methods.

| Step | Method 1 | Method 3 | Method 2 |
|---|---|---|---|
| 0 | Compute $J(u)$, $a_k$, $b_k$, and $\hat{c}$ from (11), (13), and (14) | | |
| 1 | Calculate $\{\hat{\rho}_k\}_{k=1}^{N_g}$ from (15) | | Compute $\epsilon(u)$ from (11) and (18) |
| 2 | Temporary path power: $\{\hat{p}_k\}_{k=1}^{N_g}$ from (16) | | $\hat{L}_{\max}$ by (19) and (20) |
| 3 | $\hat{L}_{\max}$ by a threshold ($\alpha$) | $\hat{L}_{\max}$ by $\Lambda_p(\cdot)$ from (23) and (24) | Calculate $\{\hat{\rho}_k\}_{N_g-\hat{d}_{\max}}^{N_g}$ by (15) and (21) |
| | $\{\hat{p}_k\}_{k=1}^{N_g}$ from (17) | $\mathbf{p}_u\big|_{u=\hat{L}_{\max}}$ in Table 1 | $\{\hat{p}_k\}_{k=0}^{L_{\max}-1}$ from (22) |
| 4 | $\hat{d}_{\max} = (\hat{L}_{\max} - 1)$, $\hat{\sigma}^2 = J(\hat{L}_{\max})$, $\mathbf{p}_{\hat{L}_{\max}}$ | | |

## 4. PDP Based MMSE Channel Estimation

Let us define the channel correlation matrix in the time-domain as

$$\mathbf{R}_{hh} = E\left[\mathbf{h}\mathbf{h}^H\right], \tag{25}$$

where $(\cdot)^H$ denotes the Hermitian transpose, and $\mathbf{h}$ is the $N_g \times 1$ channel column vector of

$$\mathbf{h}\big|_{N_g \times 1} = [h_0, \mathbf{0}, h_1, \cdots, h_{L-1}, \mathbf{0}]^T, \tag{26}$$

where the non-zero $L$ elements of $\{h_l\}_{l=0}^{L-1}$ in (4) are located at $\{d_l\}_{l=0}^{L-1}$ at $\mathbf{h}(d_l) = h_l$, and $(\cdot)^T$ denotes the transpose. Therefore, the channel correlation matrix can be obtained in the frequency-domain as

$$\mathbf{R}_{HH} = \mathbf{F}[\mathbf{R}_{hh}]_{N \times N}\mathbf{F}^H, \tag{27}$$

where $\mathbf{F}$ is the $N \times N$ DFT matrix, and $[\cdot]_{N \times N}$ denotes the operation that expands input $[\cdot]$ to a $N \times N$ matrix through zero-padding. The channel PDP is written as

$$\mathbf{p}_{L_{\max}} = \mathrm{diag}(\mathbf{R}_{hh}) = \left[\sigma_0^2, \mathbf{0}, \sigma_1^2, \cdots, \sigma_{L-1}^2, \mathbf{0}\right], \tag{28}$$

where the non-zero $L$ path powers of $\{\sigma_l^2\}_{l=0}^{L-1}$ are allocated at $\{d_l\}_{l=0}^{L-1}$ as $\mathbf{p}_{L_{\max}}(d_l) = \sigma_l^2$, and $\mathrm{diag}(\cdot)$ obtains the diagonal elements of a matrix.

Note that, without considering fractional $d_l$, $\mathbf{R}_{hh}$ is a diagonal matrix and $\mathbf{R}_{hh} \equiv \mathrm{diag}(\mathbf{p}_{L_{\max}})_{N_g \times N_g}$ where $\mathrm{diag}(\cdot)_{N_g \times N_g}$ creates a $N_g \times N_g$ diagonal matrix. On the contrary, for the fractional $d_l$ case, $\mathbf{R}_{hh}$ cannot be a diagonal matrix and $\mathbf{R}_{hh} \neq \mathrm{diag}(\mathbf{p}_{L_{\max}})_{N_g \times N_g}$. We describe this as related to the fractional $d_l$, in Section 5.

From (17), (22), and $\mathbf{p}_{\hat{L}_{\max}}$ in Table 2, the estimated channel PDP can be expressed as

$$\mathbf{p}_{\hat{L}_{\max}} = \left[\hat{p}_0, \hat{p}_1, \cdots, \hat{p}_{N_g-1}\right]. \tag{29}$$

From (25), (27), and (29), the estimated channel correlation matrices can be presented, in the time-domain and in the frequency-domain, as

$$\hat{\mathbf{R}}_{hh} = \text{diag}\left(\mathbf{p}_{\hat{L}_{\max}}\right)_{N_g \times N_g} \tag{30}$$

and

$$\hat{\mathbf{R}}_{HH} = \mathbf{F}\left[\hat{\mathbf{R}}_{hh}\right]_{N \times N} \mathbf{F}^H. \tag{31}$$

Then, we obtain the MMSE weight matrix as

$$\mathbf{W} = \hat{\mathbf{R}}_{HH}\left(\hat{\mathbf{R}}_{HH} + \hat{\sigma}^2 \mathbf{I}\right)^{-1}, \tag{32}$$

where $\hat{\sigma}^2 = J\left(\hat{L}_{\max}\right)$, and $\mathbf{I}$ denotes the $N \times N$ identity matrix. Then, the MMSE channel estimation coefficient can be expressed as

$$\mathbf{H}_{MMSE} = \mathbf{W}\mathbf{H}_{ini}, \tag{33}$$

where $\mathbf{H}_{ini}$ is the initially estimated channel gain. In general, the pilot-symbol-assisted channel estimation scheme has $\mathbf{H}_{ini} = \mathbf{H}_{LS}$. In this paper, we assume that $\mathbf{H}_{ini} = \mathbf{H}_{WSUM-TDA}$, as in [7].

## 5. Simulation Results

In this paper, we demonstrate the efficiency of the proposed channel estimation schemes through simulations based on the IEEE 802.11p standard [1,25]. The key parameters in IEEE 802.11p are shown in Table 3. The transmitter and the receiver basically adopt the convolutional encoder and the Viterbi decoder with constraint length 7, respectively [1,25]. We assume that one packet consists of 100 OFDM symbols, and the received signal is stored in the buffer in packet units. When the buffer size is $B_f$, $\mathbf{p}_{\hat{L}_{\max}}$ from (29) is estimated using the current received packet and the past $\left(B_f - 1\right)$ packets, and the total number of OFDM symbols used in the estimation process is $M = 100B_f$. We adopt QPSK with coding rate of $1/2$. For all cases, the packet error rate (PER) performance is averaged over $5 \times 10^5$ packet transmissions with SNR $= E_s\sigma_h^2/\sigma^2$.

**Table 3.** Parameters in IEEE 802.11p [9].

| Parameters | Value |
|---|---|
| Bandwidth | 10 MHz |
| Modulation order | BPSK, QPSK, 16QAM, 64QAM |
| Total no. of subcarriers (DFT size $= N$) | 64 |
| No. of data subcarriers | 48 |
| No. of pilot subcarriers | 4 |
| Pilot subcarrier index | $-21, -7, 7, 21$ |
| Sampling time $(T_s)$ | 0.1 μs |
| Guard interval $(T_{CP} = N_g T_s)$ | 1.6 μs |
| OFDM symbol duration $(T_{sym} = (N_g + N)T_s)$ | 8.0 μs |

For our simulations, we have employed the 'CohdaWireless V2V channel model' in [26]. Among the five scenarios presented in [26], we considered both 'Street Crossing NLOS with 126 km/h' and 'Highway NLOS with 252km/h', of which the channel profiles are presented in Table 4. The other parameters, such as the Doppler spectrum, for each channel tap are listed in [26]. In our simulation, we employ the fractional $d_l$ by considering $T_s = 0.1$ μs in Table 3 so that 'Street Crossing NLOS' has $d_1 \in \{2,3\}$ and $d_3 \in \{5,6\}$, and 'Highway NLOS' has $d_2 \in \{4,5\}$, as shown in Table 4. For the fractional case, the given path power is divided into two according to the relative distance to two adjacent sampling time locations [27]. Figures 3a and 4b show both $\mathbf{R}_{hh}$ and $\mathbf{p}_{L_{\max}}$ of 'Street Crossing

NLOS' and 'Highway NLOS', respectively. Note that $\mathbf{R}_{hh}$ is not a diagonal matrix, and $\mathbf{p}_{L_{\max}} = \text{diag}(\mathbf{R}_{hh})$ from (28). Figures 4 and 5 show $\mathbf{R}_{HH}$ and $\mathbf{R}_{HH}^D$ of 'Street Crossing NLOS', respectively, in which $\mathbf{R}_{HH}^D$ is given by

$$
\begin{aligned}
\mathbf{R}_{hh}^D &= \text{diag}(\mathbf{p}_{L_{\max}})_{N_g \times N_g} \\
\mathbf{R}_{HH}^D &= \mathbf{F}\left[\mathbf{R}_{hh}^D\right]_{N \times N}\mathbf{F}^H
\end{aligned}
\tag{34}
$$

In addition, Figures 6 and 7 show $\mathbf{R}_{HH}$ and $\mathbf{R}_{HH}^D$ of 'Highway NLOS', respectively. In the four figures, the values are shown for data and pilot subcarriers, except for null and DC subcarriers, among a total of $N(= 64)$ components. From Figure 3 to Figure 7, it can be said that the proposed estimation method, $\overset{\wedge}{\mathbf{R}}_{HH}$ from (31), estimates $\mathbf{R}_{HH}^D$ from (34) as shown in Figures 5 and 7 for the practical channel $\mathbf{R}_{HH}$ as shown in Figures 4 and 6.

**Table 4.** Channel profile due to scenario in [26].

| Ch. Type | Item | Tap 0 | Tap 1 | Tap 2 | Tabp3 | Units |
|---|---|---|---|---|---|---|
| Street Crossing | Power | 0 | −3 | −5 | −10 | dB |
| NLOS | Delay ($\tau_l$) | 0 | 267 | 400 | 533 | ns |
| (126 km/h, | $d_l$ | $d_0 = 0$ | $d_1 \in \{2,3\}$ | $d_2 = 4$ | $d_3 \in \{5,6\}$ | $\times T_s$ |
| $L_{\max} = 7$) | Doppler | 0 | 295 | −98 | 591 | Hz |
| Highway | Power | 0 | −2 | −5 | −7 | dB |
| NLOS | Delay ($\tau_l$) | 0 | 200 | 433 | 700 | ns |
| (252 km/h, | $d_l$ | $d_0 = 0$ | $d_1 = 2$ | $d_2 \in \{4,5\}$ | $d_3 = 7$ | $\times T_s$ |
| $L_{\max} = 8$) | Doppler | 0 | 689 | −492 | 886 | Hz |

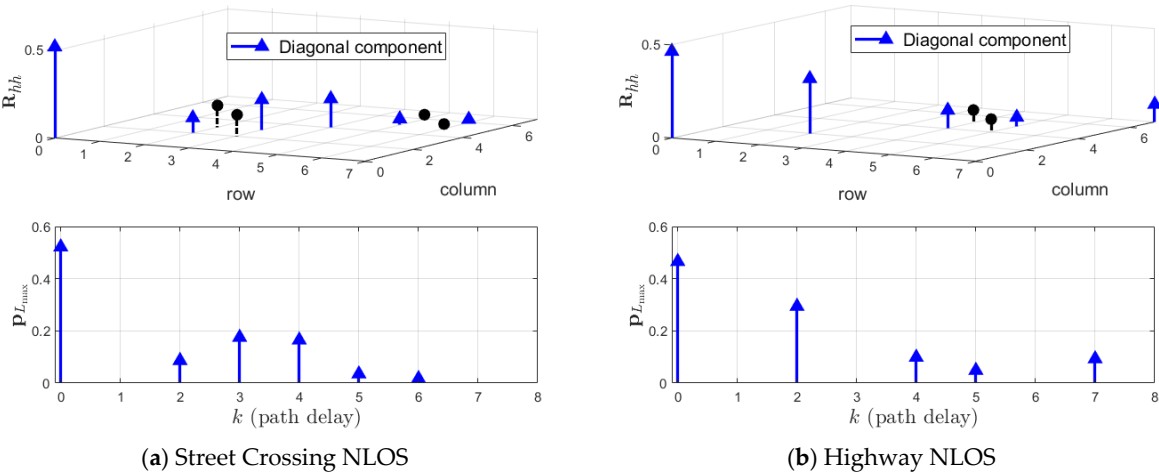

(**a**) Street Crossing NLOS　　　　　　　　　(**b**) Highway NLOS

**Figure 3.** $\mathbf{R}_{hh}$ and $\mathbf{p}_{L_{\max}}$ of Street Crossing NLOS and Highway NLOS.

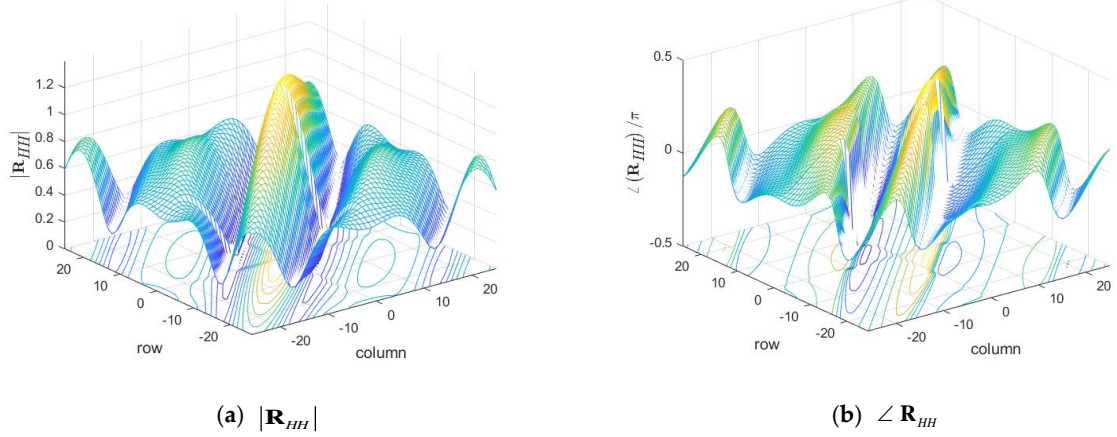

(**a**) $\left|\mathbf{R}_{HH}\right|$　　　　　　　　　　　(**b**) $\angle\,\mathbf{R}_{HH}$

**Figure 4.** $\mathbf{R}_{HH}$ of Street Crossing NLOS.

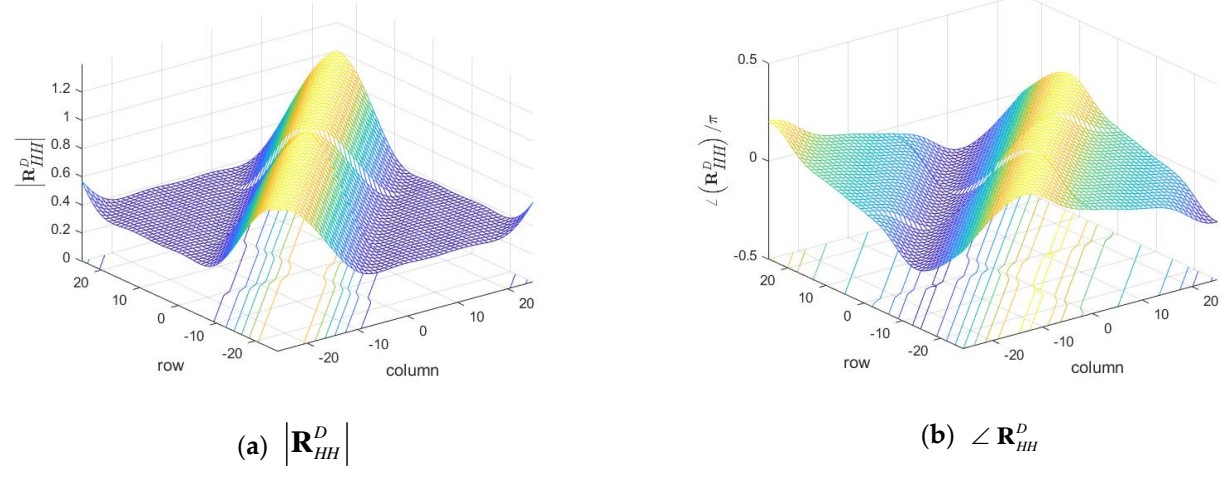

(a) $\left|\mathbf{R}_{HH}^{D}\right|$

(b) $\angle\,\mathbf{R}_{HH}^{D}$

**Figure 5.** $\mathbf{R}_{HH}^{D}$ of Street Crossing NLOS.

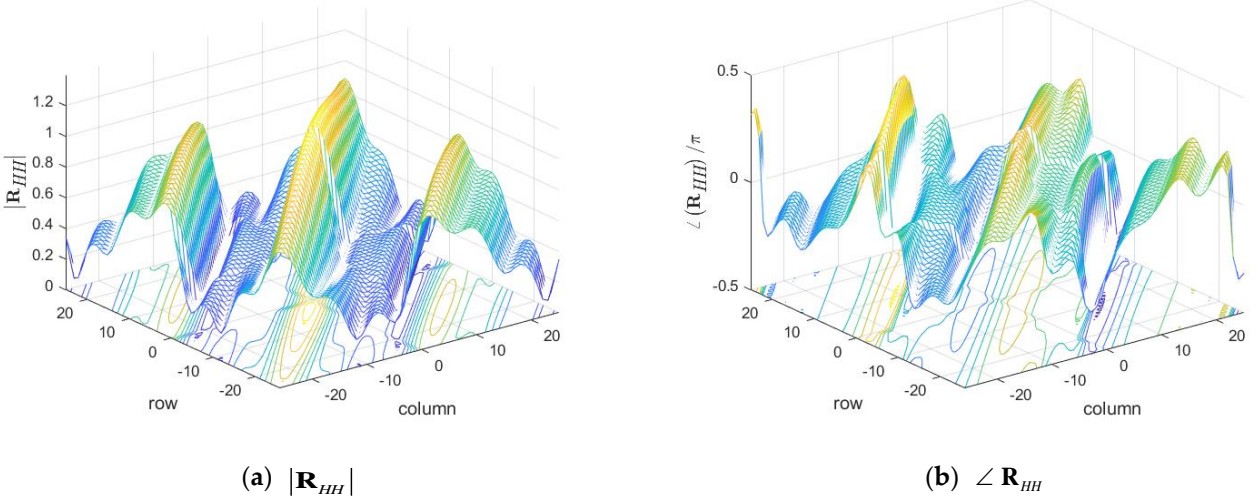

(a) $\left|\mathbf{R}_{HH}\right|$

(b) $\angle\,\mathbf{R}_{HH}$

**Figure 6.** $\mathbf{R}_{HH}$ of Highway NLOS.

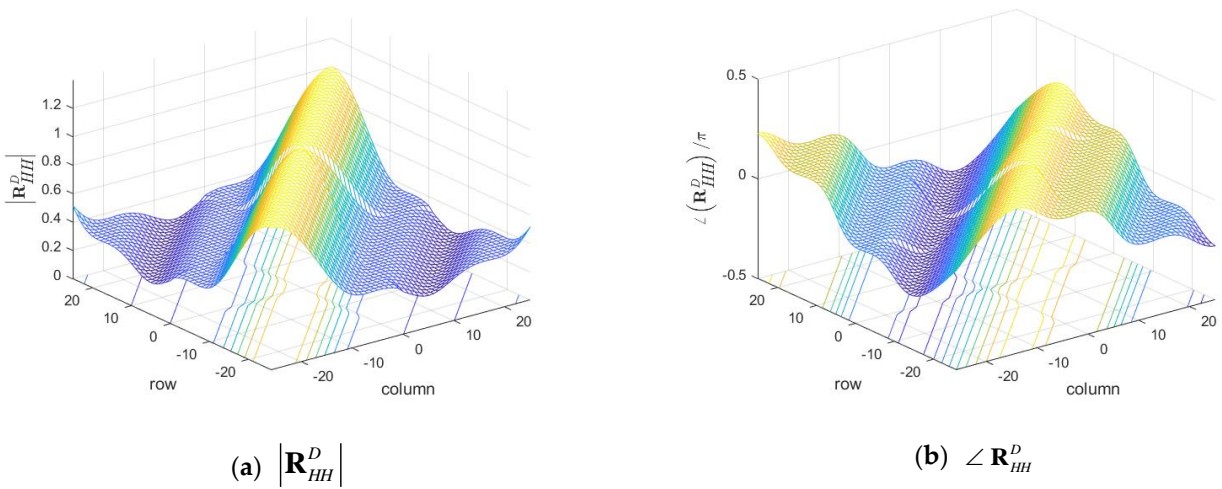

(a) $\left|\mathbf{R}_{HH}^{D}\right|$

(b) $\angle\,\mathbf{R}_{HH}^{D}$

**Figure 7.** $\mathbf{R}_{HH}^{D}$ of Highway NLOS.

*5.1. Simulation Results for PDP Estimators*

From here, let us compare the performance of the three methods in Section 3. When the NMSE of $J(u)$ in (11) is defined as $E\left[\left|J(u)-\sigma^2\right|^2\right]/\sigma^4$, Figures 1 and 2 show it for

'Street Crossing NLOS' and 'Highway NLOS', respectively. From both figures, it can be seen that there is a different trend depending on the region to which $u$ belongs. For $1 \leq u < L_{max}$, the NMSE of $J(u)$ increases with SNR because of the residual interference (i.e., inter-symbol interference (ISI)). Note that $u = L_{max}$ results in the smallest NMSE. For $L_{max} < u \leq N_g (= 16)$, the NMSE of $J(u)$ is slightly increased compared to the optimal performance of $J(L_{max})$, but it is maintained according to the SNR. Moreover, even for $J(L_{max})$, it can be seen that the NMSE slightly increases at a high SNR, which is due to the time-varying effect of the channel. As shown in Table 4, the velocity in 'Highway NLOS' is greater than that in 'Street Crossing NLOS'. Therefore, we can observe from Figures 1 and 2 that the time-varying effect of a channel is larger in 'Highway NLOS' than in 'Street Crossing NLOS'.

For $\hat{L}_{max}$, we define the correct detection (CD), the erroneous detection (ED), the bad detection (BD), and the good detection (GD) probabilities, respectively, as

$$\begin{aligned}
P_{CD} &= \Pr\{\hat{L}_{max} = L_{max}\} \\
P_{ED} &= \Pr\{\hat{L}_{max} < L_{max}\} \\
P_{BD} &= \Pr\{(L_{max} + N_g)/2 \leq \hat{L}_{max}\} \\
P_{GD} &= \Pr\{L_{max} \leq \hat{L}_{max} < (L_{max} + N_g)/2\}
\end{aligned} \tag{35}$$

In order to compare the three methods of Section 3, as shown in Figures 8 and 9, we show $P_{CD}$, $P_{ED}$, $P_{BD}$, and $P_{GD}$, for 'Street Crossing NLOS' and 'Highway NLOS' with regard to different values of $M$ and $\alpha$. When the NMSE of $\hat{\sigma}^2$ is defined as $E\left[\left|\hat{\sigma}^2 - \sigma^2\right|^2\right]/\sigma^4$, Figure 1 and Figure 10 show the NMSE of $\hat{\sigma}^2$ for 'Street Crossing NLOS' and 'Highway NLOS', respectively. From Figures 8–11, it can be seen that the performance of the PDP estimator improves as $M$ increases.

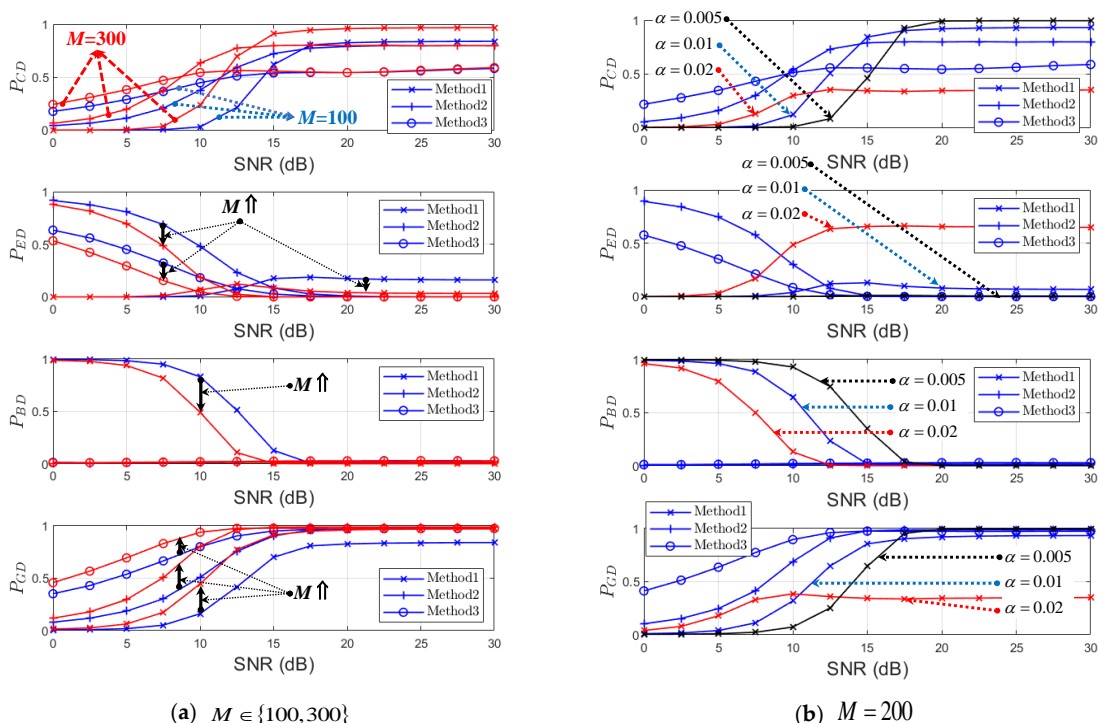

(**a**) $M \in \{100, 300\}$      (**b**) $M = 200$

**Figure 8.** Probabilities of CD, ED, BD, and GD for $\hat{L}_{max}$ (Street Crossing NLOS, $M \in \{100, 200, 300\}$, $\alpha = 0.01$ ).

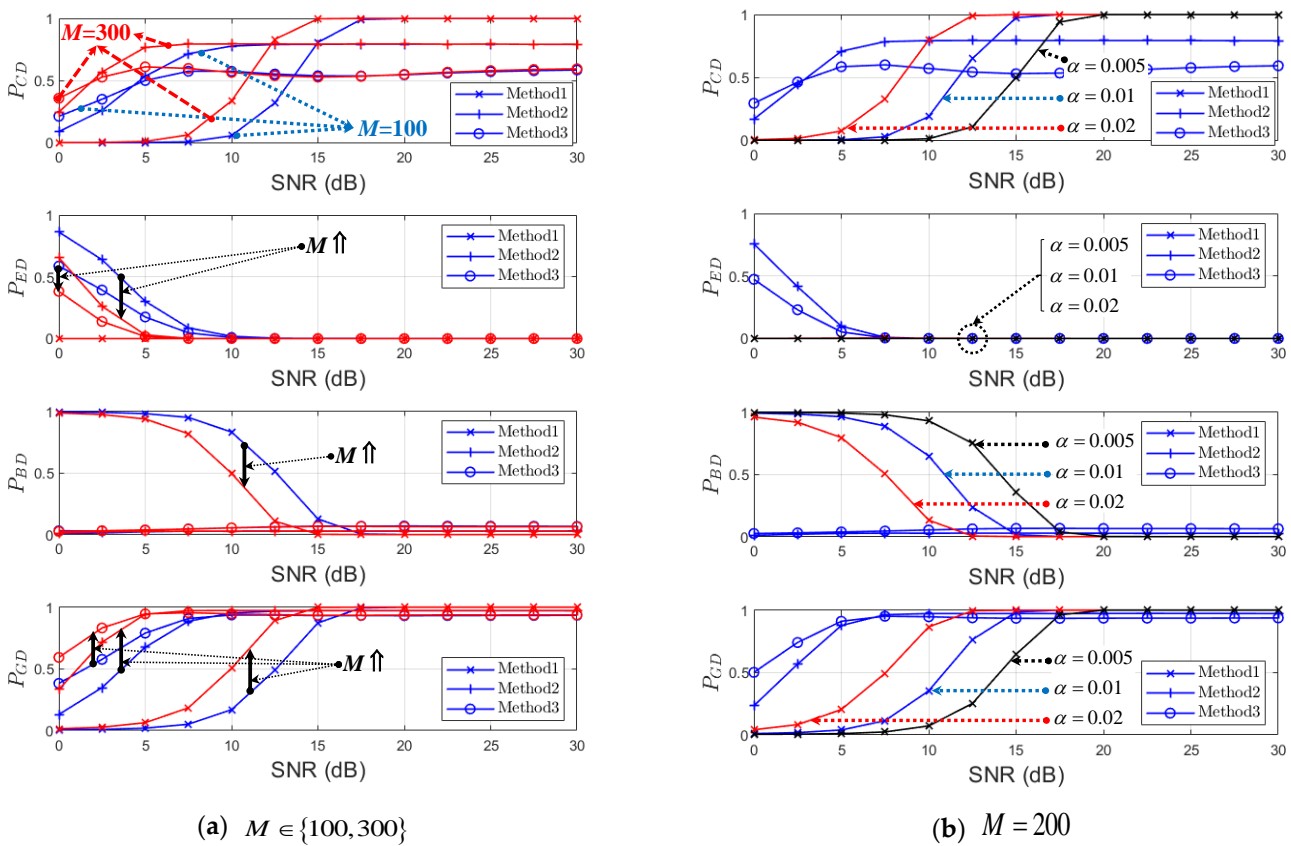

(**a**) $M \in \{100, 300\}$　　　　　　　　(**b**) $M = 200$

**Figure 9.** Probabilities of CD, ED, BD, and GD for $\hat{L}_{\max}$ (Highway NLOS, $M \in \{100, 200, 300\}$, $\alpha = 0.01$ ).

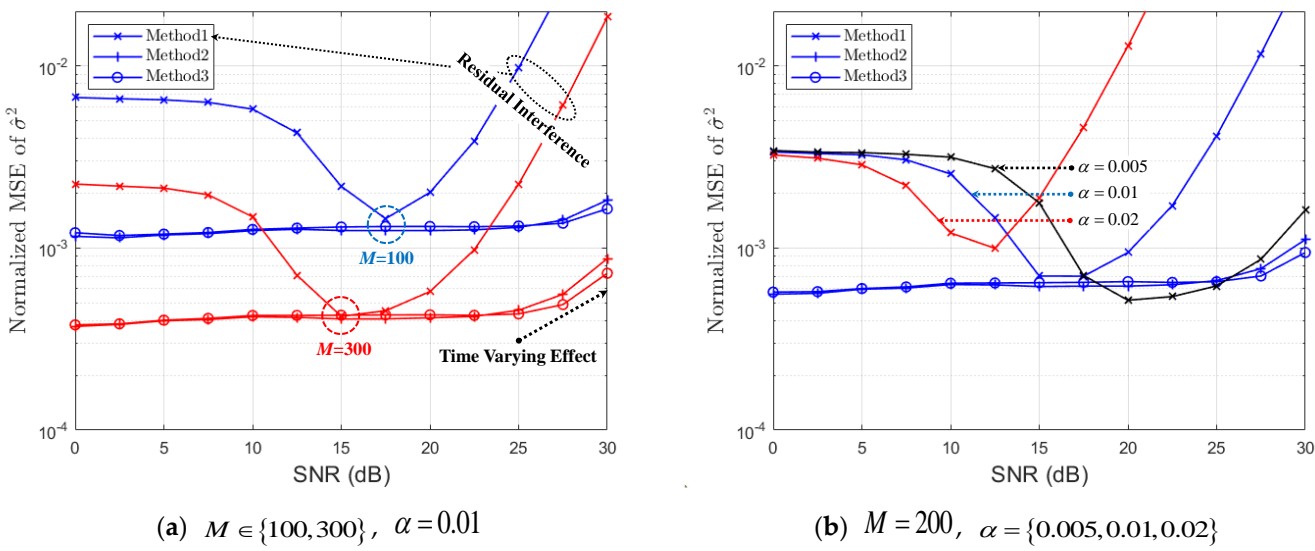

(**a**) $M \in \{100, 300\}$, $\alpha = 0.01$　　　　　　　(**b**) $M = 200$, $\alpha = \{0.005, 0.01, 0.02\}$

**Figure 10.** NMSE of $\hat{\sigma}_2$ (Street Crossing NLOS).

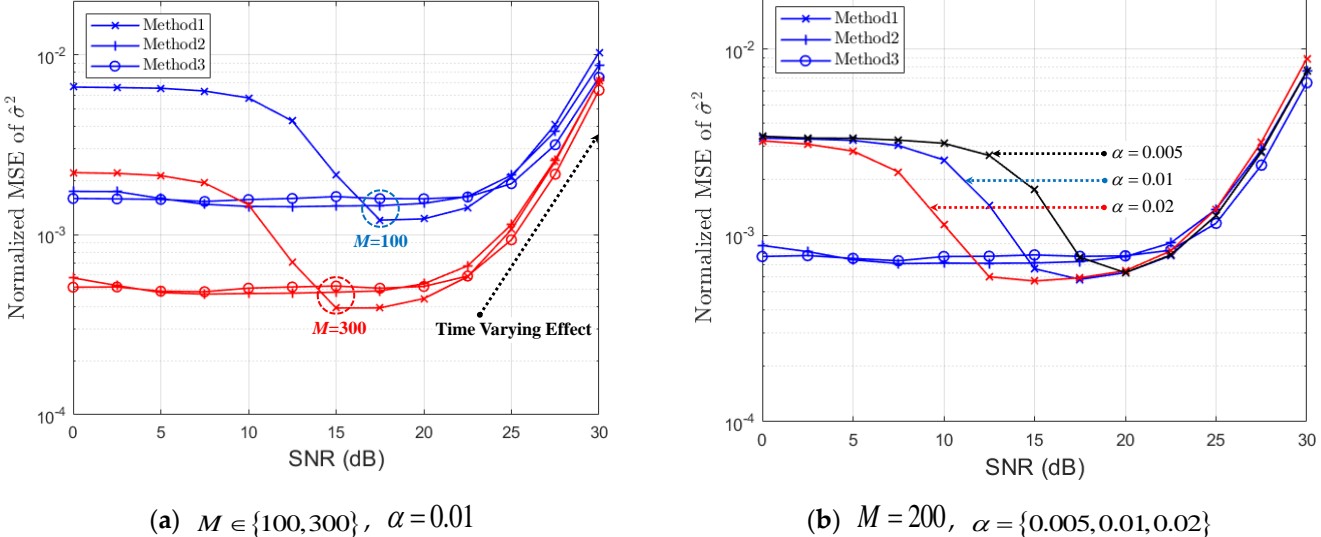

**(a)** $M \in \{100, 300\}$, $\alpha = 0.01$　　　　　　　　**(b)** $M = 200$, $\alpha = \{0.005, 0.01, 0.02\}$

**Figure 11.** NMSE of $\hat{\sigma}_2$ (Highway NLOS).

In the high SNR region in Figure 8a, 'Method 1' has a higher CD probability than other methods but a non-zero ED probability, resulting in a low GD probability. Note that a non-zero ED probability means the case of $\hat{L}_{\max} < L_{\max}$ (i.e., residual interference) at Figure 1 and the NMSE of 'Method 1' with $\alpha = \{0.01, 0.02\}$ are observed to greatly increase at Figure 10. In Figure 11 of 'Highway NLOS', this phenomenon is not observed. Therefore, we can say that the performance of 'Method 1' depends on the channel environment and $\alpha$. In the low SNR region in Figures 8 and 9, 'Method 1' has a low ED probability but a high BD (or low GD) probability, which indicates the occurrence of the case $(L_{\max} + N_g)/2 \leq \hat{L}_{\max}$ in Figures 1 and 2. Therefore, it can be observed that 'Method 1' results in a higher NMSE than other methods, in the low SNR region, as shown in Figures 10 and 11. On the contrary, 'Method 2' and 'Method 3' outperform 'Method 1' in the low SNR region with respect to the CD, BD, and GD probabilities. In the high SNR region, both 'Method 2' and 'Method 3' have generally lower CD probabilities, but lower BD probabilities and higher GD probabilities.

In general, both 'Method 2' and 'Method 3' result in higher GD probabilities for all SNR regions. This results in, as shown in Figures 10 and 11, both 'Method 2' and 'Method 3' showing stable NMSE performance in all SNR ranges.

On the other hand, the performance of 'Method 1' depends on SNR and a threshold $\alpha$. Notice that 'Method 1' tends to have better performance at a low SNR when $\alpha$ is large and better performance at a high SNR when $\alpha$ is small. As mentioned above, the NMSE increases slightly at high SNRs due to the time-varying effect of the channel. From the observation results so far, it is confirmed that both 'Method 2' and 'Method 3', without a threshold $\alpha$, show a stable performance regardless of the channel environment and SNR.

### 5.2. Simulation Results for Error Rate Performance

For a PER and BER performance comparison, we present three performance bounds which are 'Perfect CE', 'Ideal $-$ $\mathbf{R}_{hh}$', and 'Ideal $-$ $\mathbf{p}_{L\max}$', respectively. 'Perfect CE' denotes that the channel coefficient obtained by DFT on the actual time-varying channel value at the middle position of each OFDM symbol, $\left\{ h_{l,m}(N/2)\big|_{l=0}^{L-1} \right\}$, is applied. Both 'Ideal $-$ $\mathbf{R}_{hh}$' and 'Ideal $-$ $\mathbf{p}_{L\max}$' denote that $\mathbf{R}_{hh}$ from (27) and $\mathbf{p}_{L\max}$ from (28) are assumed to be known to the receiver, and then, $\mathbf{W} = \mathbf{R}_{HH}\left(\mathbf{R}_{HH} + \sigma^2\mathbf{I}\right)^{-1}$ with (27) and $\mathbf{W} = \mathbf{R}_{HH}^D\left(\mathbf{R}_{HH}^D + \sigma^2\mathbf{I}\right)^{-1}$ with (34) are used, respectively, at the receiver with $\mathbf{H}_{ini} = \mathbf{H}_{WSUM-TDA}$ in [7]. Notice that, as mentioned before, for the fractional $d_l$ case, $\mathbf{R}_{hh} \neq \mathbf{R}_{hh}^D = \text{diag}\left(\mathbf{p}_{L\max}\right)_{N_g \times N_g}$, so that 'Ideal $-$ $\mathbf{p}_{L\max}$' denotes the achievable performance bound of the proposed three PDP-based MMSE schemes, which are expressed as 'Prop–M.1', 'Prop–M.2', and 'Prop–M.3'.

Figures 12 and 13 show a PER and BER performance comparison with respect to channel estimation schemes under 'Street Crossing NLOS with 126 km/h' for QPSK, with $M \in \{100, 300\}$, and $M = 200$, respectively. Figures 14 and 15 show a PER and BER performance comparison with respect to channel estimation schemes under 'Highway NLOS with 252 km/h' for QPSK, with $M \in \{100, 300\}$ and $M = 200$, respectively. Note that a threshold $\alpha = 0.01$ is used for 'Prop–M.1' in Figures 12 and 14, and Figures 13 and 15 show the performance comparison for 'Prop–M.1' with respect to threshold $\alpha$.

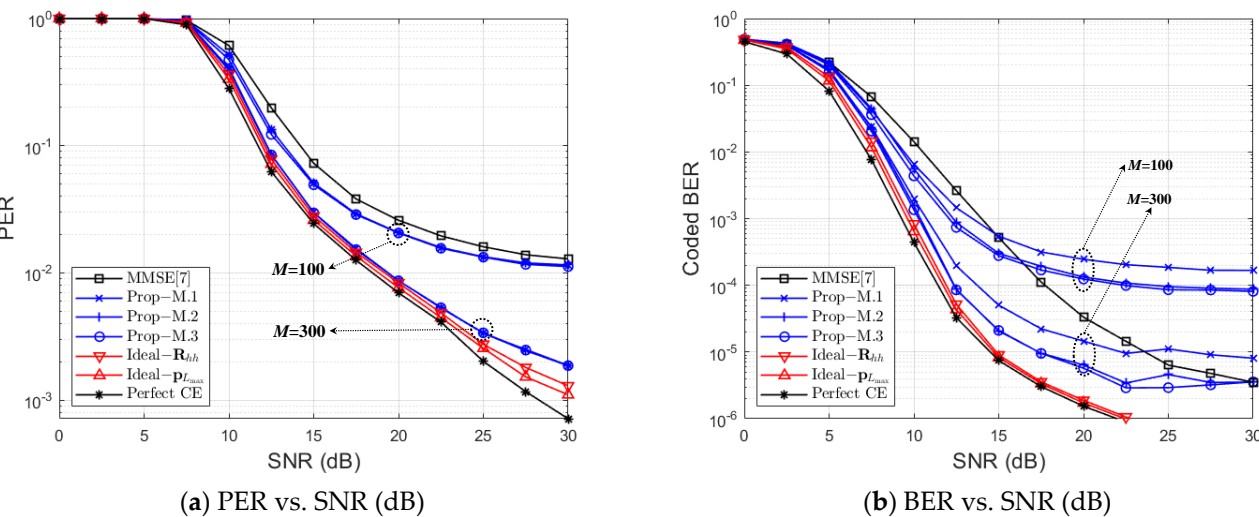

**(a)** PER vs. SNR (dB)　　　　　　　**(b)** BER vs. SNR (dB)

**Figure 12.** Error Performance Comparison at Street Crossing NLOS (QPSK, CR = 1/2, $M \in \{100, 300\}$, $\alpha = 0.01$).

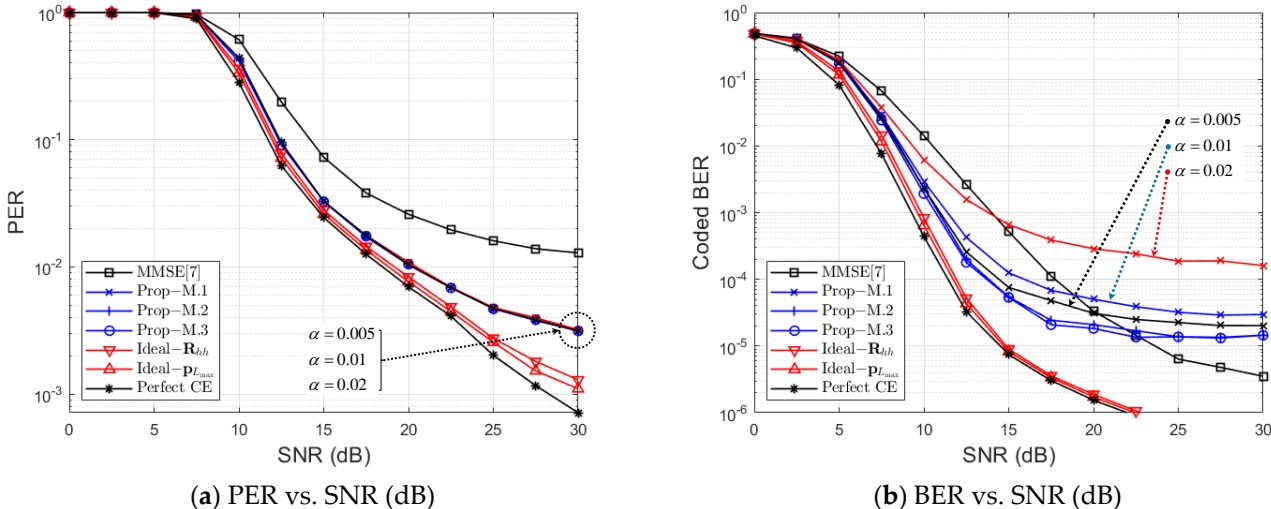

**(a)** PER vs. SNR (dB)　　　　　　　**(b)** BER vs. SNR (dB)

**Figure 13.** Error Performance Comparison at Street Crossing NLOS (QPSK, CR = 1/2, $M = 200$, $\alpha = \{0.005, 0.01, 0.02\}$).

First, let us compare three performance bounds for PER and BER. From Figures 12–15, 'Perfect CE' shows the lowest error rate bound, and 'Ideal $- \mathbf{R}_{hh}$' and 'Ideal $- \mathbf{p}_{L_{\max}}$' both show a similar performance to 'Perfect CE'. Therefore, we can say that, even for the fractional $d_l$ case, the achievable performance bound of the MMSE-CE method can be obtained by using the proposed PDP-based MMSE-CE methods. Furthermore, it can be confirmed that the proposed three methods can approach the PER performance bound of 'Ideal $- \mathbf{p}_{L_{\max}}$' as $M$ increases.

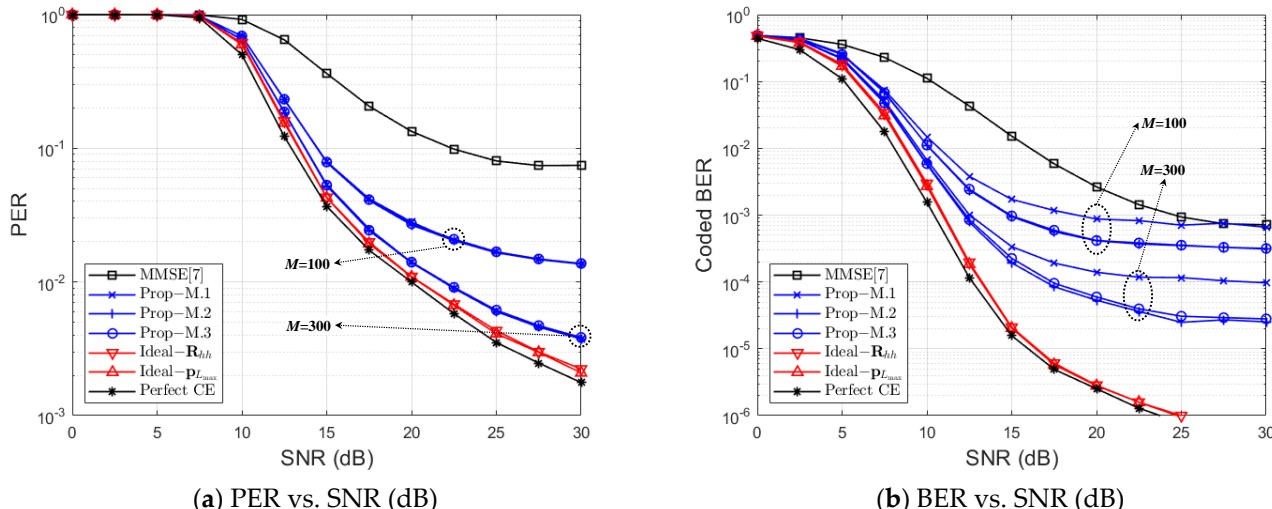

(**a**) PER vs. SNR (dB)　　　　　　(**b**) BER vs. SNR (dB)

**Figure 14.** Error Performance Comparison at Highway NLOS (QPSK, CR = 1/2, $M \in \{100, 300\}$, $\alpha = 0.01$ ).

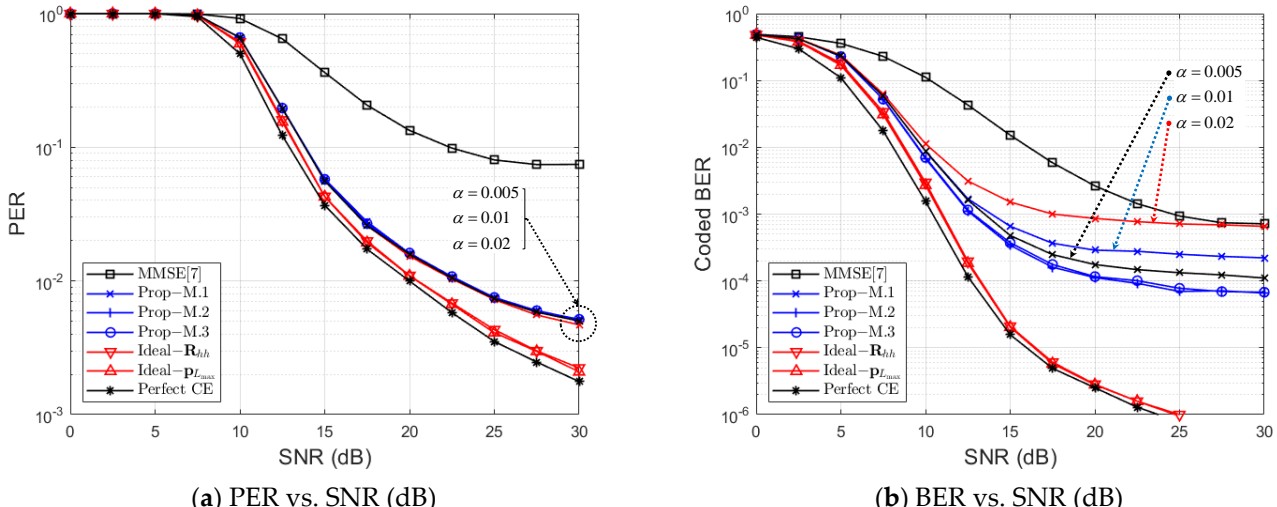

(**a**) PER vs. SNR (dB)　　　　　　(**b**) BER vs. SNR (dB)

**Figure 15.** Error Performance Comparison at Highway NLOS (QPSK, CR = 1/2, $M = 200$, $\alpha = \{0.005, 0.01, 0.02\}$ ).

In four figures, 'MMSE [7]' indicates the MMSE-CE scheme of [7], in which an inverse matrix operation is performed for every OFDM symbol. From Figures 12–15, it is verified that the proposed three schemes show better PER performance than 'MMSE [7]' at all SNR regions and all $M$. The proposed schemes show better BER performance than 'MMSE [7]' in a specific SNR region in 'Street Crossing NLOS', as shown at both Figures 12b and 13b, and in all SNR regions in 'Highway NLOS', as shown at both Figures 14b and 15b. Even though 'Prop–M.1', 'Prop–M.2', and 'Prop–M.3' show similar PER performance, 'Prop-M.1' shows a higher BER than both 'Prop–M.2' and 'Prop–M.3' as shown from Figures 12b, 13, 14 and 15b. Moreover, as shown in Figures 13b and 15b, the BER performance of 'Prop–M.1' approaches both 'Prop–M.2' and 'Prop–M.3' as the threshold $\alpha$ decreases. This is because, when the threshold $\alpha$ is reduced in 'Method 1', both the ED and GD performance are improved, as shown in Figures 8b and 9b, and the noise variance estimation performance is improved, as shown in Figures 10b and 11b. Furthermore, from Figures 12 and 15, it is shown that the proposed methods can achieve PER= $10^{-2}$ at a reasonable SNR.

## 6. Conclusions

In this paper, we presented the MMSE channel estimation schemes for OFDM systems with three types of PDP estimators. Among the PDP estimators, the first is a threshold-based method in [17], and the second is described in [18] with our modification. The last one in [19] is the method with a structure without a threshold value, similar to the second method. Numerical simulations indicate the robustness of the PDP estimators in [18,19] according to the SNR and channel environment. Furthermore, through simulations over time-varying correlated fading channels, the PDP-based MMSE channel-estimation schemes can be used to obtain the performance close to the achievable bounds. In particular, it was confirmed that 0.01 PER can be obtained with 1 dB SNR loss at $M = 200$, without SNR loss at $M = 300$ for 'Street Crossing NLOS', with 2.5 dB SNR loss at $M = 200$, and with 1.5 dB SNR loss at $M = 300$ for 'Highway NLOS', respectively.

**Author Contributions:** Conceptualization, S.L. and K.K.; methodology, H.W.; software, S.L.; validation, H.W., S.L., and K.K.; formal analysis, H.W.; investigation, S.L.; resources, H.W.; data curation, S.L.; writing—original draft preparation, S.L.; writing—review and editing, H.W. and K.K.; visualization, S.L.; supervision, H.W.; project administration, K.K.; funding acquisition, H.W., S.L. and K.K. All authors have read and agreed to the published version of the manuscript.

**Funding:** This work was supported by the National Research Foundation of Korea (NRF) grant funded by the Korean Government (MSIT) (NRF-2020R1A2C1005260, NRF-2021R1A2C2012558, NRF-2021R1A2C1014063).

**Institutional Review Board Statement:** Not applicable.

**Informed Consent Statement:** Not applicable.

**Data Availability Statement:** Not applicable.

**Conflicts of Interest:** The authors declare no conflict of interest.

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
