# Peer review of "Power-Delay-Profile-Based MMSE Channel Estimations for OFDM Systems"

_electronics, doi:10.3390/electronics12030510_

Round 1

Reviewer 1 Report

1) What is the computational complexity of the proposed algorithms?

2) How can the authors compare the complexity of their algorithms with the ones they have modified?

3) How do the authors compute PER?

4) What is the relationship between PER and BER?

5) Which coding scheme has been used in simulations? The authors mention QPSK with a coding rate of 1/2. Do they mean convolutional code? If yes, what is the reason/justification for using this code?

Author Response

Thank you for your constructive suggestion. We checked them according to the reviewer’s comment and changed them.

Reviewer 2 Report

Review of the paper "Power Delay Profile Based MMSE Channel Estimations for OFDM systems", by Sungmook Lim, Hanho Wang and Kyunbyoung Ko.

The paper examines channel estimation in OFDM over time-varying fading channels, using cyclic prefix in systems with insufficient pilot/training symbols. The goal is to implement an MMSE estimation scheme based on estimated power delay profile.

The paper first reviews the available literature on the topic, with emphasis on references 17-19. However, the overall feel is that the paper is written more as a review article (or even a part of a dissertation), rather than a research article. Namely:

Section 2
equations (1-5) are almost identical to [17, (1-5)],
(7) is [17 (15)]
(8) is [17 (7)]
(9, 10) are [17 (17, 18)]

Subsection 3.1
equations (12-16) are mostly identical to [17, (9-13)]

Subsection 3.2
(18) is an unnumbered equation from [18] (between equations (3) and (4) therein)
(19, 20) are [18 (4, 5)]

Subsection 3.3
I could not find the reference, but from the structure of the paper, it is possible that a number of equations are present in both papers.

There is no need to reiterate the already known equations, unless it is the case for just a few of them and they are essential for the flow of the paper. In this paper, my opinion is that the number of non-essential equations that are repeated is large. The authors should restructure the paper so that repeating these equations is not necessary, and referencing the source papers should be enough.

Figures 6 and 7 are the first ones to be referenced in the paper  at the start of section 3, so the numbering of the figures should be reconsidered. Figures should be numbered according to the order of referencing.

The conclusions are supported by the simulation results presented, but the number of figures illustrating theses conclusions seems large. It is my opinion that the number of figures could be reduced to show the most important results and focus more closely on the contribution of the paper.

Author Response

We appreciate your valuable advice. Based on your recommendation, we have revised the earlier manuscript to response reviewers’ comments clearly.

Reviewer 3 Report

The paper is interesting and it is well explained.

I have only two suggestions

The introduction should be enhanced with specific latest literatures from Power delay profiles based MSME channel estimations.

Conclusion should be written with numerical findings.

Author Response

(The authors gave the same response as above.)

Reviewer 4 Report

In this manuscript, the MMSE channel estimation schemes for OFDM 367
systems with three type PDP estimators were presented. In my point of view, the contributions are interesting and the paper was described in a good manner. the results are sufficient.

Author Response

We appreciate your valuable advice. Based on your recommendation, we have revised the earlier manuscript to response reviewers’ comments clearly. As a result, we could improve the revised manuscript based on your advice and reviewer’s comments.

Round 2

Reviewer 1 Report

The authors have addressed all my comments. The paper can be accepted for publication.